# Performance Evaluation of Combined Hydrothermal-Mechanical Pretreatment of Lignocellulosic Biomass for Enzymatic Enhancement

**DOI:** 10.3390/polym14122313

**Published:** 2022-06-08

**Authors:** Jiraporn Phojaroen, Thitirat Jiradechakorn, Suchata Kirdponpattara, Malinee Sriariyanun, Jatupol Junthip, Santi Chuetor

**Affiliations:** 1Department of Chemical Engineering, Faculty of Engineering, King Mongkut’s University of Technology North Bangkok (KMUTNB), Bangkok 10800, Thailand; s6301031810028@email.kmutnb.ac.th (J.P.); s6301031820066@email.kmutnb.ac.th (T.J.); suchata.k@eng.kmutnb.ac.th (S.K.); 2Biorefinery and Process Automation Engineering Centre (BPAEC), King Mongkut’s University of Technology North Bangkok (KMUTNB), Bangkok 10800, Thailand; macintous@gmail.com; 3The Sirindhorn International Thai-German Graduate School of Engineering, King Mongkut’s University of Technology North Bangkok (KMUTNB), Bangkok 10800, Thailand; 4Faculty of Science and Technology, Nakhon Ratchasima Rajabhat University, Nakhon Ratchasima 30000, Thailand; jatupol.j@nrru.ac.th

**Keywords:** hydrothermal-mechanical pretreatment, agricultural wastes, energy efficiency, waste generation

## Abstract

Pretreatment is a crucial process in a lignocellulosic biorefinery. Corncob is typically considered as a natural renewable carbon source to produce various bio-based products. This study aimed to evaluate the performance of the hydrothermal-mechanical pretreatment of corncob for biofuels and biochemical production. Corncob was first pretreated by liquid hot water (LHW) at different temperatures (140–180 °C) and duration (30, 60 min) and then subjected to centrifugal milling to produce bio-powders. To evaluate the performance of this combined pretreatment, the energy efficiency and waste generation were investigated. The results indicated that the maximum fermentable sugars (FS) were 0.488 g/g biomass obtained by LHW at 180 °C, 30 min. In order to evaluate the performance of this combined pretreatment, the energy efficiency and waste generation were 28.3 g of FS/kWh and 7.21 kg of waste/kg FS, respectively. These obtained results indicate that the combined hydrothermal-mechanical pretreatment was an effective pretreatment process to provide high energy efficiency and low waste generation to produce biofuels. In addition, the energy efficiency and waste generation will be useful indicators for process scaling-up into the industrial scale. This combined pretreatment could be a promising pretreatment technology for the production of biofuels and biochemicals from lignocellulosic valorization.

## 1. Introduction

The amount of energy requirement has increased as part of a rapidly growing population, which directly causes several problems in terms of environmental sustainability and an economic crisis. The deficiency in conventional energy resources such as petroleum and natural gases has been a struggle for the sustainable development goals.

Several proven alternative energy resources have been developed to partially replace the main energy resources. Lignocellulosic biomass represents its potential for producing various value-added products including biofuels, biochemicals, and biomaterials via a biorefinery [1,2,3,4]. The utilization of lignocellulosic material as a renewable carbon source for high value-added products has been developed over the last decade [5]. The lignocellulosic biomass is typically composed of three major natural biomolecules including cellulose hemicellulose and lignin, which are associated together with complex chemical bonding to form a recalcitrant structure [6,7]. This structural complexity and recalcitrance make the proper lignocellulosic valorization become more difficult and have a low conversion yield [8,9,10]. However, the scientific breakthrough and research on lignocellulosic valorization have been emphasized to be more applicable on a large scale in order to reduce the environmental concerns. To maximize the use of lignocellulosic biomass, one of the most important processes in a biorefinery is the pretreatment process, which is responsible for altering and deconstructing the complex structure for the release of active components to be converted into active component biomolecules for further applications. The pretreatment process mostly affects the physical and chemical barriers of the lignocellulosic structure for its enhancement [11,12].

The existing pretreatment technologies have been divided into four categories including physical, chemical, physicochemical, and biological pretreatment. The physical pretreatment consists of alleviating the physical properties (crystallinity, degree of polymerization, and porosity) of the material by using several technologies such as mechanical size reduction and irradiation. The chemical pretreatment is mostly the use of acid and alkaline pretreatments, which are typically responsible for breaking down the chemical bonding in the lignocellulosic structure. Moreover, ionic liquid and deep eutectic solvents (DESs) have recently been developed for lignocellulosic pretreatment, which significantly enhances the delignification. The physicochemical pretreatment mostly consists of the steam explosion and liquid hot water, and the biological pretreatment consists of mainly using the enzymes of fungi for the removal of lignin [1,7].

However, these pretreatment technologies possess drawbacks such as high energy consumption, undesired waste generation, high installation costs, and long duration. The pretreatment process basically counts for 40% of the total capital cost of lignocellulosic valorization because of the high energy consumption, high chemical usage, and undesired waste generation [2,13]. Even though the existing pretreatment technologies lead to the enhanced utilization of lignocellulosic material, the single pretreatment process is inadequate to cover all of the biorefinery processes in terms of the technological, economic, and environmental considerations. Moreover, the existing biorefinery process is still developing because of the high investment cost, in particular, the pretreatment process, which generally counts for 30–40% of the total investment cost.

The recent challenges in developing the proper biorefinery processes include the reduction in investment costs, no undesired waste generation, no inhibitors for biological conversion, and a comparable production yield. Several studies have revealed that the combination of pretreatment has been effective in various aspects such as high active component recovery, less waste generation, economic feasibility, and environmentally-friendly processes.

Hydrothermal pretreatment presents an industrial potential for integration into the biorefinery process. It consists of hemicellulose solubilization and lignin removal via the autohydrolysis reaction of water at high temperature and pressure, and in particular, this process does not require the addition and recovery of chemicals different from water, and it can be said that hydrothermal processes are environmentally-friendly fragmentation processes [14,15]. Autohydrolysis is used to pretreat the lignocellulosic material to disrupt the linkage between cellulose-hemicellulose and lignin, which could enhance the enzymatic hydrolysis. Several cellulosic enzymes such as cellulase provide an effective saccharification, in particular, cellulose [16].

The hydrothermal pretreatment consists of using the liquid state of water at high pressure and temperature to form an H_3_O^+^ for hemicellulose solubilization and partial lignin removal [17]. The hydrothermal pretreatment is typically operated at an elevated temperature between 140–200 °C and high pressure in the range of 10–20 bar in a Parr reactor [18,19]. A number of studies on hydrothermal pretreatment indicated its efficiency to solubilize the hemicellulose component by approximately 60–80% compared to the untreated samples [20,21]. Furthermore, the hydrothermal pretreatment, along with the chemical catalysts, could achieve a high lignin removal for further enzymatic accessibility.

On the other hand, one of the most efficient physical pretreatments is mechanical size reduction, which can tackle the physical properties including cellulose crystallinity and the degree of polymerization of lignocellulosic material for further processes [22,23,24]. However, the mechanical size reduction process typically needs a high energy input to break down the internal structure, and this technology has its own limitations with regard to integration into the biorefinery process.

To overcome these aforementioned limitations of pretreatment technology, a combination of hydrothermal and mechanical pretreatment is an effective and promising strategy in an integrated biorefinery for the production of biofuels, biochemicals, and biomaterials [25]. The hydrothermal pretreatment, followed by mechanical size reduction, leads to the solubilization of the hemicellulose and partial lignin removal for enzymatic enhancement as well as making the material softer and more porous. Moreover, the combined process affects the crystallinity of the cellulose to be more amorphous, thus increasing the reactivity of the particles in subsequent processing [26]. This current study investigated the effect of a combined hydrothermal-mechanical pretreatment of corncob on its performance on the production of fermentable sugars. The combined process was evaluated in terms of energy efficiency and waste generation compared to its efficiency in enhancing the production yield of fermentable sugars. Two established indicators, energy efficiency and waste generation, can be used as the crucial factor for the further techno-economic analysis of the process. The process integration could define the holistic bioethanol production from corncob material. This research could lead to further development, application management, and valorization of agricultural byproducts to create a bio-circular economy and to decrease CO_2_ emissions, which would consequently slow down global warming.

## 2. Materials and Methods

### 2.1. Raw Material Preparation

Corncob (CC) was obtained from farmers in Maetha District, Lampang Province, Thailand. Samples were oven-dried until the moisture content in the samples was approximately 8–10%. Samples were coarsely ground by knife milling to obtain the particle size of approximately 2–4 mm and subjected to a sieve with a screen size of 1 mm. The samples were stocked in a Ziploc plastic bag at room temperature for the further pretreatment process.

### 2.2. Hydrothermal Pretreatment

CC was pretreated by using a hydrothermal technique without a chemical catalyst. A sample of 30 g CC was placed in a stainless-steel Parr reactor 1 L. This pretreatment was conducted by using distilled water boiled under a pressure of 20 bar and a temperature between 140–180 °C for two different durations (30 and 60 min) and a rotation speed of 200 rpm [26]. The solid fraction was collected and washed with distilled water for neutralization and oven-dried at 60 °C to obtain a constant weight. The dried CC was stocked at room temperature for the mechanical size reduction process.

### 2.3. Mechanical Pretreatment

In this study, the CC samples hydrothermally pretreated at three different temperatures and two different durations were used to conduct the mechanical size reduction. The CC samples were ground by using centrifugal milling using a 0.25 mm screen size. The centrifugal milling was equipped with a digital wattmeter to measure the intensity and voltage during the grinding process.

In the case of the specific energy consumption calculation due to the mechanical size reduction, this can be solved with Equation (1)
(1)$$E_{sp}=\int_{t_{o}}^{t}\left(P_{t}-P_{t_{o}}\right)dt/m$$
where *E_sp_* is the total specific energy consumption (kWh/kg); *P_t_* is the power in watt consumed at time *t*; *P_t0_* is the average power consumption in Watt under empty conditions (without biomass); and m is the mass in kg of material to be ground.

### 2.4. Monosaccharide Concentration Analysis

The monosaccharide concentration including glucose, xylose, and arabinose of the biomass was determined using the standard method for biomass analysis provided by the National Renewable Energy Laboratory (NREL), Golden, CO, USA [27]. The concentration of monosaccharides such as glucose and xylose in the soluble fraction was measured using high-performance liquid chromatography (CTO-10AS VP, Shimadzu, Kyoto, Japan) equipped with an Aminex HPX-87 H column (Bio-Rad Laboratories, Inc., Hercules, CA, USA). The column temperature was 65 °C with 0.005 M of sulfuric acid as the mobile phase at a flow rate of 0.5 mL/min. Monosaccharides of an analytical grade with a known concentration were used as the standards [12]. 

### 2.5. Enzymatic Saccharification

Enzymatic hydrolysis of the untreated and pretreated CC was performed by using commercial enzymes Cellulast CTec2 (Novozymes, Belgrave, Denmark). The reaction (5 mL total volume) contained 5% of the solid biomass sample (on a dry weight basis) with 10 FPU/g enzyme loading in 50 mM of sodium acetate buffer, pH 5 adjusted with acetic acid. Sodium azide was added at the end of the experiment to inhibit microbial growth. The reaction was incubated at 50 °C for 72 h with 200 rpm agitation [12]. The experiment was performed in triplicate. The amount of monosaccharide concentration was quantified by HPLC as described above. Moreover, the enzymatic efficiency was calculated as:(2)$$\mathrm{Enzymatic}\;\mathrm{Efficiency}\;\left(\%\right)=\frac{The\;amount\;of\;glucose\;\left(\mathrm{kg}\right)}{The\;amount\;of\;cellulose\;\left(\mathrm{kg}\right)}*100$$

### 2.6. Energy Efficiency and Waste Generation Evaluation

Energy efficiency was used to evaluate the performance of an integrated hydrothermal-mechanical pretreatment. It was defined by the ratio of output and input, where the input is the total energy consumption and the output is the total fermentable sugars released by enzymatic hydrolysis. The energy efficiency was calculated according to Chuetor et al. [12] as:(3)$$\mathrm{Energy}\;\mathrm{Efficiency}\left(\frac{\mathrm{kg}}{\mathrm{kWh}}\right)=\frac{Total\;amount\;of\;fermetable\;sugars\;\left(\mathrm{kg}\right)}{Total\;energy\;consumption\;\left(\mathrm{kWh}\right)}$$

Waste generation was used to investigate the undesired waste generated during the pretreatment process, which was defined by the ratio of total waste generation and the total amount of fermentable sugars obtained from enzymatic hydrolysis. The waste generation was calculated as:(4)$$\mathrm{Waste}\;\mathrm{generation}\;\left(\frac{\mathrm{kg}}{\mathrm{kg}}\right)=\frac{Total\;amount\;ofwaste\;generation\;\left(\mathrm{kg}\right)}{Total\;amount\;of\;fermentable\;sugars\;\left(\mathrm{kg}\right)}$$
where the total waste is calculated by the difference between the total reactant (biomass + water) and the total fermentable sugars.

## 3. Results and Discussion

### 3.1. Effect of Temperature and Duration on Enzymatic Hydrolysis Efficiency

Hydrothermal pretreatment of CC was conducted at different temperatures ranging from 140–180 °C with two different durations (30 and 60 min). Figure 1 illustrates the variation of the biochemical compositions, cellulose, hemicellulose, lignin, and others of each sample. It was noticed that the cellulose content relatively increased with the increasing temperature whereas the hemicellulose content decreased in a solid fraction. The maximum cellulose content was 52.90%, obtained by CC 180 °C at 60 min, which corresponded to a 100% increase compared to the native CC. Furthermore, Figure 1 indicates that the hemicellulose content trended to decrease significantly because of its solubilization during hydrothermal pretreatment. Imman et al. revealed that the hydrothermal pretreatment of corncob was effectively caused by the solubilization of hemicellulose, which was due to the hydronium ions being responsible for breaking down the cellulose–hemicellulose–lignin linkages [28,29]. Concerning the lignin content, it still remained in the solid fraction and was hard to remove during hydrothermal pretreatment, which would certainly restrict the enzymatic hydrolysis of CC due to the lignin content and its derived inhibition impact [30,31].

From the aforementioned results in the biochemical composition, it interestingly noticed that the hydrothermal pretreatment was effective for hemicellulose solubilization and cellulose enrichment, which was subsequently favorable for further enzymatic hydrolysis [32,33]. Figure 1 also shows that the enzymatic efficiency increased with the temperature. The highest enzymatic efficiency was 75.30% obtained by CC at 180 °C at 30 min, which corresponded to a 216% increase compared to a native CC. This large augmentation of enzymatic hydrolysis efficiency was due to the hemicellulose solubilization during hydrothermal pretreatment [34].

### 3.2. Evolution of Energy Consumption during the Combined Pretreatment Process

In this study, the CC was pretreated hydrothermally in the Parr reactor at three different temperatures (145, 165, and 180 °C) and two different durations (30 and 60 min), followed by mechanical size refining, and evaluated for energy consumption and enzymatic hydrolysis. The combined hydrothermal-mechanical pretreatment led to a decrease in the particle size, which subsequently increased the specific surface area for further enzymatic digestibility. Table 1 shows the amount of energy used in each process of the combined pretreatment method. The total energy consumption during the hydrothermal pretreatment increased when the temperature increased, while the energy consumption during the grinding process decreased because the CC structure had been destroyed due to the hydrothermal pretreatment process before.

The highest total energy consumption was obtained from CC pretreated for 30 min at 180 °C, which corresponded to 17.8 kWh/kg of biomass. It was approximately 66% of the addition of energy compared to the lowest total energy consumption that was obtained by CC pretreated for 60 min at 165 °C, which corresponded to 10.7 kWh/kg of biomass. Interestingly, the essential energy consumption was due to the energy-consuming during hydrothermal pretreatment, which was in the range of 6.5–16.5 kWh/kg biomass. This indicates that the energy consumption of the Parr reactor increased with the increased temperature. Concerning the energy consumption due to the mechanical size reduction, the energy consumption decreased with the increased temperature that caused the structural changes during pretreatment [35,36].

### 3.3. Effect of Pretreatments on Enzymatic Hydrolysis

The effect of pretreatment duration on the enzymatic hydrolysis was evaluated. In this study, three different temperatures and two different durations were used to investigate the evolution of fermentable sugar concentration after pretreatment. The total fermentable sugar content was calculated to evaluate the performance of a combined hydrothermal-mechanical process. Figure 2 illustrates that when the temperature increased, the amount of fermentable sugar increased (T_180 °C_ > T_165 °C_ > T_140 °C_), respectively. In the case of pretreatment duration, the longer the duration in the reactor, the more fermentable sugars were released (D_60 min_ > D_30 min_). These indicate that the temperature and duration of the pretreatment affected the internal structural deconstruction of the corncob to enzyme accessibility.

The highest fermentable sugar was 0.488 kg/kg of biomass obtained from CC pretreated at 180 °C for 30 min, which corresponded to a 356.07% increase in the fermentable sugar compared to the control (0.107 kg/kg of biomass). This increase in fermentable sugar was due to the structural deconstruction of the biomass and the removal of the hemicellulose and lignin. The high amount of fermentable sugars obtained by enzymatic hydrolysis was related to the enzymatic efficiency, as seen in Figure 1. Several studies have suggested that the augmentation of fermentable sugar concentration was due to the structural alteration through the combined pretreatment process, which is typically an important stage for lignocellulosic valorization.

### 3.4. Evaluation of Energy Efficiency and Waste Generation of Combined Hydrothermal-Mechanical Pretreatment

To evaluate the performance of the combined hydrothermal-mechanical pretreatment, the relationship between the total energy consumption and total fermentable sugars was evaluated through the estimation of the energy efficiency. The energy efficiency was used to evaluate the performance of this developed combined pretreatment to provide fermentable sugars. Energy efficiency is defined as the ratio of the total product (fermentable sugars) to the total energy consumption consumed during pretreatment. Table 2 indicates that the highest energy efficiency was obtained by the CC pretreated at 165 °C for 60 min, which corresponded to 0.041 kg of product/kWh. The effect of temperature on energy efficiency showed that the elevated temperature provided the high energy efficiency that was due to the high fermentable sugars obtained from enzymatic hydrolysis. On the other hand, the effect of pretreatment duration revealed that the longer duration released more fermentable sugars. These obtained results were relatively associated with the obtained results of the biochemical compositions and enzymatic efficiency, as seen in Table 1.

Beyond sustainability, the combined hydrothermal-mechanical pretreatment has also been investigated for waste generation during pretreatment. The waste generation represents the total amount of waste generated during the pretreatment process, which is calculated from the ratio of the total waste generated and total fermentable sugars. The obtained results showed that the minimum waste generation was 7.21 kg of waste/kg product obtained from the CC pretreated at 180 °C at 30 min, as seen in Table 2. The waste generation was the total waste generated in solid and liquid form in the process. To compare different pretreatment conditions, the waste generation was varied between 7.2–26.7 kg of waste/kg of fermentable sugars.

## 4. Conclusions

The development of an economically environmentally-friendly pretreatment technology for integration into a biorefinery is important for industrial-scale production. The combined hydrothermal-mechanical pretreatment is potentially an alternative solution process for lignocellulosic valorization. The integrated hydrothermal and mechanical pretreatment could achieve the structural alteration of lignocellulosic material for enzymatic enhancement. The combined pretreatment provided a high fermentable sugar concentration compared to the untreated material. The maximum fermentable sugar was 0.488 kg/kg of biomass obtained at 180 °C at 30 min, which was due to the hemicellulose solubilization via the hydrothermal pretreatment and the increase in the specific surface area via the mechanical size reduction. The energy efficiency was used to evaluate the performance of the combined pretreatment, which could be used for further economic analysis. The highest energy efficiency was 0.0283 kg of fermentable sugars/kWh. On the other hand, the waste generation was investigated in terms of the environmental impacts of the process. The lowest waste generation was 7.21 kg of waste/kg product obtained at 180 °C at 30 min. The aforementioned results showed that the combined hydrothermal-mechanical pretreatment provided a high fermentable sugar concentration, which consequently enhanced further biofuel production.

These obtained results suggest that the combination of hydrothermal-mechanical pretreatment could be a promising pretreatment technology in terms of energy efficiency and an environmentally-friendly process for lignocellulosic valorization.

## Figures and Tables

**Figure 1 polymers-14-02313-f001:**
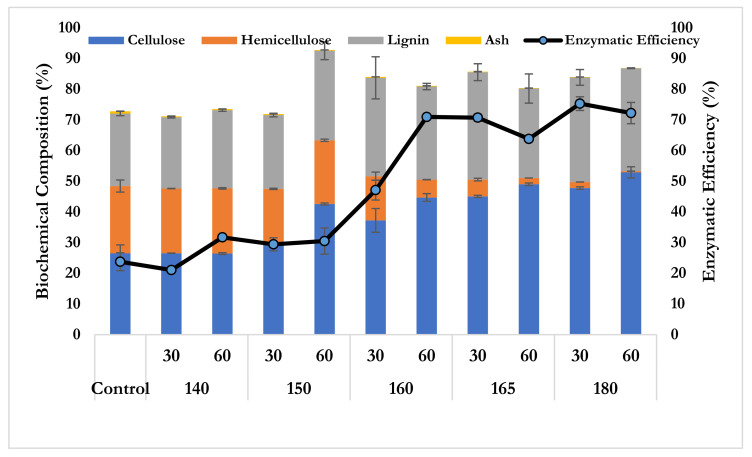
The biochemical composition and enzymatic hydrolysis efficiency of corncob pretreated by the hydrothermal pretreatment.

**Figure 2 polymers-14-02313-f002:**
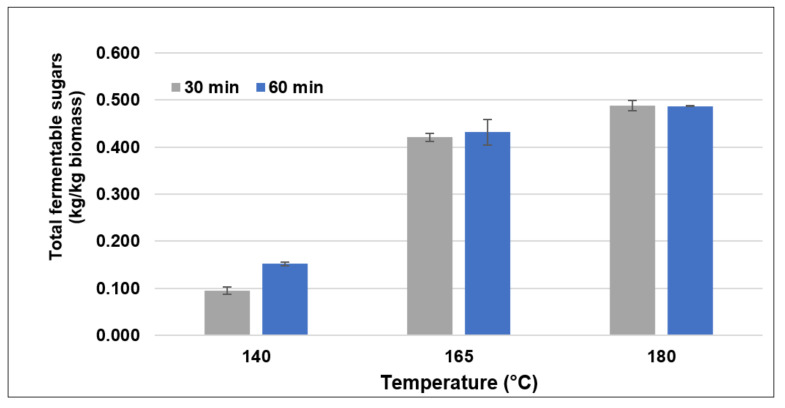
The total fermentable sugars from the enzymatic hydrolysis of different pretreatment conditions.

**Table 1 polymers-14-02313-t001:** The energy consumption of corncob pretreated by combined pretreatment at different conditions.

Temperature (°C)	Time (min)	E Mechanical (kWh/kg)	E LHW (kWh/kg)	E Drying (kWh/kg)
140	30 60	0.532 0.321	6.628 6.534	0.649 0.650
165	30 60	0.126 0.102	9.964 9.972	0.650 0.652
180	30 60	0.033 0.025	16.567 16.639	0.659 0.660

**Table 2 polymers-14-02313-t002:** The comparison of the energy efficiency and waste generation at different pretreatment conditions.

Temperature (°C)	Reducing Sugars (kg/kg Biomass)		Total Energy Consumption (kWh/kg Biomass)		Energy Efficiency (kg FS/kWh)		Waste Generation (kg Waste/kg Product)	
	30 min	60 min	30 min	60 min	30 min	60 min	30 min	60 min
140	0.095	0.152	17.808	17.505	0.0055	0.0088	26.71	15.34
165	0.421	0.432	10.740	10.726	0.0396	0.0406	8.42	8.9
180	0.488	0.487	17.259	17.324	0.0283	0.0282	7.21	7.23

## Data Availability

This study did not report any data.

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
