# Peer review of "Performance Evaluation of Combined Hydrothermal-Mechanical Pretreatment of Lignocellulosic Biomass for Enzymatic Enhancement"

_polymers, 2022, doi:10.3390/polym14122313_

Round 1
Reviewer 1 Report
This is a manuscript on the applied polymer technology. It has fuzzy introduction, fuzzy abstract, fuzzy conclusions. The introduction section is not a review of the literature. The authors should briefly formulate the problem which will be solved in the work.
The abstract is not specific, see e.g. lines 25-27.
Integrals (1,2) should be omitted, they are not used. Hot water at 140-180 deg C (line 122) immediately requires comments.
Abbreviation FSC (line 23) is not good.
Other minor comments.
Line 95. ?
Lines 136, 137. W should be lower case.
Lines 210, 217, W should be capital.
Author Response
We are thankful to the reviewers of the journal for spending their valuable time reviewing the manuscript.
We have carefully addressed all the comments and have revised the manuscript accordingly.
The modifications done in the manuscript are summarized below in our responses and the changes can be tracked in the manuscript.
Responses:
This is a manuscript on applied polymer technology. It has fuzzy introduction, fuzzy abstract, fuzzy conclusions. The introduction section is not a review of the literature. The authors should briefly formulate the problem which will be solved in the work.
- The abstract is not specific, see e.g. lines 25-27.
Ans: Revised as the suggestion in line 27-28.
- Integrals (1,2) should be omitted, they are not used.
Ans: Revised as the suggestion but reserved the equation 2 for understanding how we calculate the energy consumption due to mechanical size reduction.
- Hot water at 140-180°C (line 122) immediately requires comments.
Ans: This temperature range was obtained from several articles such as
https://doi.org/10.1016/j.biortech.2021.126024
https://doi.org/10.1016/j.biortech.2021.126033
https://doi.org/10.1016/j.biortech.2021.126075
and added a reference in line 135.
- Abbreviation FSC (line 23) is not good
Ans: Revised and changed it to FS.
- Line 95?
Ans: Revised as the suggestion
- Lines 136, 137. W should be lower case.
Ans: Revised as the suggestion
- Lines 210, 217, W should be capital.
Ans: Revised as the suggestion

Reviewer 2 Report
In this paper, the author uses the combination of hydrothermal pretreatment and mechanical and treatment to reduce the energy input in the process of biological refining. More energy-saving and environment-friendly pretreatment methods will drive the further development of biomass fuels. However, there are still some issues to be addressed. A moderate revision is suggested before its acceptance.
1. In order to better reflect the advantages of the combination of hydrothermal pretreatment and mechanical pretreatment, the author needs to introduce the four pretreatment methods respectively, provide references for readers, and describe their advantages and disadvantages.
2. Please carefully check the spelling and wording of the full text and correct relevant errors (such as the beginning of line 95).
3. More introduction on the biorefinery of lignocellulose should be provided with supporting highly relevant papers: Integrated lignocellulosic biorefinery: 1) Gateway for production of second generation ethanol and value added products; 2) Opportunities for New Biorefinery Products from Ethiopian Ginning Industry By-products: Current Status and Prospects
4. Please pay attention to the serial number of subheadings to avoid low-level errors (Part II).
5. All test methods not proposed by the author need to provide systematic references and standard sources (such as test and analysis of monosaccharide concentration).
6. For the component content of cellulose, hemicellulose and lignin mentioned below, the test and analysis scheme shall be given above.
7. Enzyme pyrosis is reported for the wood treatment to produce cellulose-derived materials. Necessary introduction should be performed to show the differences between the reported work and this work, and further to present the novelty of this work. Please carefully read and cite: Pyrolysis of Enzymolysis-Treated Wood: Hierarchically Assembled Porous Carbon Electrode for Advanced Energy Storage Devices; Enzymolysis-treated wood-derived hierarchical porous carbon for fluorescence-functionalized phase change materials
8. The sentence for “The utilization of lignocellulosic material as a renewable carbon source for high value-added products has been developed last decade year” should be supported with article: Conversion of biomass lignin to high-value polyurethane: A review
9. The definition of production waste and its general composition need to be clarified.
10. In the analysis, the author rashly put forward the enzymatic hydrolysis efficiency, which needs to give a specific definition and calculation method in the previous test and analysis.
11. The legend of Figure 1 is incomplete. Please complete the legend of the gray part of the histogram.
12. The figure used in the article is too brief. Please draw it with special software to improve professionalism.
13. Please elaborate the influence mechanism of hydrothermal pretreatment and mechanical pretreatment on the highest fermentable sugar yield.
Author Response
We are thankful to the reviewers of the journal for spending their valuable time reviewing the manuscript.
We have carefully addressed all the comments and have revised the manuscript accordingly.
The modifications done in the manuscript are summarized below in our responses and the changes can be tracked in the manuscript.
Responses:
In this paper, the author uses the combination of hydrothermal pretreatment and mechanical and treatment to reduce the energy input in the process of biological refining. More energy-saving and environment-friendly pretreatment methods will drive the further development of biomass fuels. However, there are still some issues to be addressed. A moderate revision is suggested before its acceptance.
- In order to better reflect the advantages of the combination of hydrothermal pretreatment and mechanical pretreatment, the author needs to introduce the four pretreatment methods respectively, provide references for readers, and describe their advantages and disadvantages.
Ans: Revised as the suggestion, we revised and added more information in line 57-68.
- Please carefully check the spelling and wording of the full text and correct relevant errors (such as the beginning of line 95).
Ans: Revised as the suggestion
- More introduction on the biorefinery of lignocellulose should be provided with supporting highly relevant papers: Integrated lignocellulosic biorefinery: 1) Gateway for production of second-generation ethanol and value-added products; 2) Opportunities for New Biorefinery Products from Ethiopian Ginning Industry By-products: Current Status and Prospects
Ans: Revised as the suggestion and added more references N° 3-4.
- Please pay attention to the serial number of subheadings to avoid low-level errors (Part II).
Ans: Revised as the suggestion
- All test methods not proposed by the author need to provide systematic references and standard sources (such as test and analysis of monosaccharide concentration).
Ans: Revised as the suggestion and added more references in part materials & methods in line 158 and 165.
- For the component content of cellulose, hemicellulose and lignin mentioned below, the test and analysis scheme shall be given above.
Ans: Revised as the suggestion and added more references in part materials & methods
- Enzyme pyrosis is reported for the wood treatment to produce cellulose-derived materials. Necessary introduction should be performed to show the differences between the reported work and this work, and further to present the novelty of this work. Please carefully read and cite: Pyrolysis of Enzymolysis-Treated Wood: Hierarchically Assembled Porous Carbon Electrode for Advanced Energy Storage Devices; Enzymolysis-treated wood-derived hierarchical porous carbon for fluorescence-functionalized phase change materials
Ans: Revised as the suggestion and added more references in line 89-90.
- The sentence for “The utilization of lignocellulosic material as a renewable carbon source for high value-added products has been developed last decade year” should be supported with article: Conversion of biomass lignin to high-value polyurethane: A review
Ans: Revised as the suggestion and added more references in line 43 reference °5.
- The definition of production waste and its general composition need to be clarified.
Ans: Revised as the suggestion in line 179-180.
- In the analysis, the author rashly put forward the enzymatic hydrolysis efficiency, which needs to give a specific definition and calculation method in the previous test and analysis.
Ans: Revised as the suggestion in line 167-169.
- The legend of Figure 1 is incomplete. Please complete the legend of the gray part of the histogram.
Ans: Revised as the suggestion in figure 1
- The figure used in the article is too brief. Please draw it with special software to improve professionalism.
Ans: Revised as the suggestion
- Please elaborate the influence mechanism of hydrothermal pretreatment and mechanical pretreatment on the highest fermentable sugar yield.
Ans: Revised as the suggestion line 288-289.

Round 2
Reviewer 2 Report
Authors have addressed all the issues well. An acceptance is suggested.
Author Response
The authors would like to thank you for your valuable suggestions to improve our manuscript.